# Expectation Complete Graph Representations Using Graph Homomorphisms

**Pascal Welke**[*]
University of Bonn, Germany
welke@cs.uni-bonn.de

**Maximilian Thiessen**[*]
TU Wien, Austria
maximilian.thiessen@tuwien.ac.at

**Thomas Gärtner**
TU Wien, Austria
thomas.gaertner@tuwien.ac.at

## Abstract

We propose and study a practical graph embedding that *in expectation* is able to distinguish all non-isomorphic graphs and can be computed in polynomial time. The embedding is based on Lovász' characterisation of graph isomorphism through an infinite dimensional vector of homomorphism counts. Recent work has studied the expressiveness of graph embeddings by comparing their ability to distinguish graphs to that of the Weisfeiler-Leman hierarchy. While previous methods have either limited expressiveness or are computationally impractical, we devise efficient sampling-based alternatives that are maximally expressive in expectation. We empirically evaluate our proposed embeddings and show competitive results on several benchmark graph learning tasks.

## 1 Introduction

We study novel efficient and expressive graph embeddings based on Lovász' characterisation of graph isomorphism through homomorphism counts. While most practical graph embeddings drop the property of *completeness*, that is, the ability to distinguish all non-isomorphic graphs, in favour of runtime, we devise efficient embeddings that retain completeness *in expectation*. To achieve that, we sample pattern graphs in a particular way, simultaneously guaranteeing completeness and polynomial runtime in expectation. We discuss related work, in particular the relationship to the $k$-dimensional Weisfeiler Leman isomorphism test, and show first results on benchmark datasets.

While subgraph counts are also a reasonable choice for expectation complete graph embeddings, they have multiple drawbacks compared to homomorphism counts. Most importantly, from a computational perspective, computing subgraph counts even for simple graphs such as trees or paths is NP-hard [Alon et al., 1995; Marx and Pilipczuk, 2014], while we can compute homomorphism counts efficiently [Díaz et al., 2002] as long as the pattern graphs have small *treewidth*, a measure of 'tree-likeness'. In particular, all known exact algorithms for subgraph isomorphism have a runtime exponentially in the pattern size or the maximum degree of the pattern even for small treewidth — one of the main reasons why the graphlet kernel [Shervashidze et al., 2009] and similar fixed pattern based approaches [Bouritsas et al., 2022] only count subgraphs up to size around 5.

Probably most important from a conceptual perspective is the relationship of homomorphism counts to the *cut distance* [Borgs et al., 2006; Lovász, 2012]. The cut distance is a well-studied and important distance on graphs that captures global structural but also sampling-based local information. It is well known that the distance given by (potentially approximated and sampled) homomorphism counts is close to the cut distance and hence has similar favourable properties. The cut distance, and hence, homomorphism counts, capture the behaviour of all permutation-invariant functions on graphs. For

---

[*]Equal contribution.

, Expectation Complete Graph Representations Using Graph Homomorphisms (Extended Abstract). Presented at the First Learning on Graphs Conference (LoG 2022), Virtual Event, December 9–12, 2022.

an ongoing discussion about the importance of the cut distance and homomorphism counts in the context of graph learning, see Dell et al. [2018], Grohe [2020], and Hoang and Maehara [2020].

Completeness in expectation essentially implies one powerful fact that no deterministic embedding with bounded expressiveness can guarantee: repetition will make the embedding more expressive eventually. If the graph embedding is complete in expectation it is guaranteed that sampling more patterns will eventually increase its expressiveness.

## 2 Complete Graph Embeddings

The graph isomorphism problem is a classical problem in graph theory and its computational complexity is a major open problem [Babai, 2016]. Following the classical result of Lovász [1967], two graphs are isomorphic if and only if they have the same infinite dimensional homomorphism count vectors. This provides a powerful graph embedding for graph classification tasks [Barceló et al., 2021; Dell et al., 2018; Hoang and Maehara, 2020].

A *graph* $G = (V(G), E(G))$ consists of a set $V(G)$ of *vertices* and a set $E(G) = \{e \subseteq V \mid |e| = 2\}$ of *edges*. The *size* of a graph is the number of its vertices. In the following $F$ and $G$ denote graphs, where $F$ represents a *pattern* graph and $G$ a graph in our training set. A *homomorphism* $\Phi : V(F) \to V(G)$ is a map that respects edges, i.e. $\{v, w\} \in E(F) \Rightarrow \{\Phi(v), \Phi(w)\} \in E(G)$. An *isomorphism* is a bijective homomorphism whose inverse is also a homomorphism. We say that a distribution $\mathcal{D}$ over a countable domain $\mathcal{X}$ has *full support* if each $x \in \mathcal{X}$ has nonzero probability.

Let $\mathcal{G}_n$ be the set of all finite graphs of size at most $n$ and let $\hom(F, G)$ denote the number of homomorphisms of $F$ to $G$ for arbitrarily graphs and $\varphi_n(G) = \hom(\mathcal{G}_n, G) = (\hom(F, G))_{F \in \mathcal{G}_n}$ denote the Lovász vector of $G$ for $\mathcal{G}_n$. Lovász [1967] proved the following classical theorem.

**Theorem 1** (Lovász [1967][2]). *Two arbitrary graphs $G, H \in \mathcal{G}_n$ are isomorphic iff $\varphi_n(G) = \varphi_n(H)$.*

We can define a simple kernel on $\mathcal{G}_n$ with the canonical inner product using $\varphi_n$.

**Definition 2** (Complete Lovász kernel). *Let $k_{\varphi_n}(G, H) = \langle \varphi_n(G), \varphi_n(H) \rangle$.*

Note that $k_{\varphi_n}$ is a *complete* graph kernel [Gärtner et al., 2003] on $\mathcal{G}_n$, i.e., $k_{\varphi_n}$ can be used to distinguish non-isomorphic graphs of size $n$. Similarly, we define complete graph embeddings.

**Definition 3.** *Let $\varphi : \mathcal{G} \to V$ be a permutation-invariant graph embedding from a family of graphs $\mathcal{G}$ to a vector space $V$. We call $\varphi$ complete (on $\mathcal{G}$) if $\varphi(G) \neq \varphi(H)$ for all non-isomorphic $G, H \in \mathcal{G}$.*

When studying graph embeddings and graph kernels we face the tradeoff between efficiency and expressiveness: complete graph representations are unlikely to be computable in polynomial time [Gärtner et al., 2003] and hence most practical graph representations drop completeness in favour of polynomial runtime. In our work, we study random graph representations. While dropping completeness and being efficiently computable, this allows us to keep a slightly weaker yet desirable property: *completeness in expectation*.

**Definition 4.** *A graph embedding $\varphi_X$, which depends on a random variable $X$, is complete in expectation if the graph embedding given by the expectation, $\mathbb{E}_X[\varphi_X(\cdot)]$, is complete.*

Similarly, we say that the corresponding kernel $k_X(G, H) = \langle \varphi_X(G), \varphi_X(H) \rangle$ is complete in expectation. We can use Lovász' isomorphism theorem to devise graph embeddings that are complete in expectation. For that let $e_F \in \mathbb{R}^{\mathcal{G}_n}$ be the '$F$th' standard basis unit-vector of $\mathcal{G}_n$.

**Theorem 5.** *Let $\mathcal{D}$ be a distribution on $\mathcal{G}_n$ with full support and $G \in \mathcal{G}_n$. Then the graph embedding $\varphi_F(G) = \hom(F, G)e_F$ with $F \sim \mathcal{D}$ and the corresponding kernel $k$ are complete in expectation.*

### 2.1 Expectation Complete Embeddings and Kernels on $\mathcal{G}_\infty$

In this section, we generalise the previous result to the set of all finite graphs $\mathcal{G}_\infty$. Theorem 1 holds for $G, H \in \mathcal{G}_\infty$ and the mapping $\varphi_\infty$ that maps each $G \in \mathcal{G}_\infty$ to an infinite-dimensional vector. The resulting vector space, however, is not a Hilbert space with the usual inner product. To see this, consider any graph $G$ that has at least one edge. Then $\hom(P_n, G) \geq 2$ for every path $P_n$ of length $n \in \mathbb{N}$. Thus, the inner product $\langle \varphi_\infty(G), \varphi_\infty(G) \rangle$ is not finite.

---

[2]see also the proof of Theorem 5.29 and the comments below in Lovász [2012].

To define a kernel on $\mathcal{G}_\infty$ without fixing a maximum size of graphs, i.e., restricting to $\mathcal{G}_n$ for some $n \in \mathbb{N}$, we define the countable-dimensional vector $\overline{\varphi}_\infty(G) = \left(\hom_{|V(G)|}(F, G)\right)_{F \in \mathcal{G}_\infty}$ where

$$\hom_{|V(G)|}(F, G) = \begin{cases} \hom(F, G) & \text{if } |V(F)| \leq |V(G)|, \\ 0 & \text{if } |V(F)| > |V(G)|. \end{cases}$$

That is, $\overline{\varphi}_\infty(G)$ is the projection of $\varphi_\infty(G)$ to the subspace that gives us the homomorphism counts for all graphs of *size at most of* $G$. Note that this is a well-defined map of graphs to a subspace of the $\ell^2$ space, i.e., sequences $(x_i)_i$ over $\mathbb{R}$ with $\sum_i |x_i|^2 < \infty$. Hence, the kernel given by the canonical inner product $\overline{k}_\infty(G, H) = \langle \overline{\varphi}_\infty(G), \overline{\varphi}_\infty(H) \rangle$ is finite and positive semi-definite. Note that we can rewrite $\overline{k}_\infty(G, H) = k_{\min}(G, H) = \langle \varphi_{n'}(G), \varphi_{n'}(H) \rangle$ where $n' = \min\{|V(G)|, |V(H)|\}$. While the first hunch might be to count patterns up to $\max\{|V(G)|, |V(H)|\}$, this is not necessary to guarantee completeness. Furthermore, the corresponding map $k_{\max}$ is not even positive semi-definite.

**Lemma 6.** $k_{\min}$ *is a complete kernel on* $\mathcal{G}_\infty$.

Given a sample of graphs $S$, we note that for $n = \max_{G \in S} |V(G)|$ we only need to consider patterns up to size $n$.[3] As the number of graphs of a given size $n$ are superexponential it is impractical to compute all such counts. Hence, we propose to resort to sampling.

**Theorem 7.** *Let $\mathcal{D}$ be a distribution on $\mathcal{G}_\infty$ with full support and $G \in \mathcal{G}_\infty$. Then $\overline{\varphi}_F(G) = \hom_{|V(G)|}(F, G)e_F$ with $F \sim \mathcal{D}$ and the corresponding kernel are complete in expectation.*

### 2.2 Sampling multiple patterns

Sampling just one pattern $F$ will not result in a practical graph embedding. Thus, we propose to sample $\ell$ patterns $F_1, \ldots, F_\ell \sim \mathcal{D}$ i.i.d. and construct the embedding $\varphi^\ell(G) \in \mathbb{N}_0^\ell$ with $(\varphi^\ell(G))_i = \hom(F_i, G)$ if $|V(F_i)| \leq |V(G)|$ and 0 otherwise for all $i \in [\ell]$. For the dot product it holds that $\varphi^\ell(G)^T \varphi^\ell(H) = \sum_{i=1}^\ell \langle \overline{\varphi}_{F_i}(G), \overline{\varphi}_{F_i}(H) \rangle$ as long as we do not sample patterns twice.[4]

## 3 Computing Embeddings in Expected Polynomial Time

A graph embedding that is complete in expectation must be efficiently computable to be practical. In this section, we describe our main result achieving polynomial runtime in expectation. The best known algorithms [Díaz et al., 2002] to exactly compute $\hom(F, G)$ take time

$$\mathcal{O}(|V(F)||V(G)|^{\text{tw}(F)+1}) \tag{1}$$

where $\text{tw}(F)$ is the *treewidth* of the pattern graph $H$. Thus, a straightforward sampling strategy to achieve polynomial runtime in expectation is to give decreasing probability mass to patterns with higher treewidth. Unfortunately, in the case of $\mathcal{G}_\infty$, this is not possible.

**Proposition 8.** *There exists no distribution $\mathcal{D}$ with full support on $\mathcal{G}_\infty$ such that the expected runtime of Eq. (1) becomes polynomial in $|V(G)|$ for all $G \in \mathcal{G}_\infty$.*

To resolve this issue we have to take the size of the largest graph in our sample into account. For a given sample $S \subseteq \mathcal{G}_n$ of graphs, where $n$ is the maximum number of vertices in $S$, we can construct simple distributions achieving polynomial time in expectation.

**Theorem 9.** *There exists a distribution $\mathcal{D}$ such that computing the expectation complete graph embedding $\overline{\varphi}_X(G)$ takes polynomial time in $|V(G)|$ in expectation for all $G \in \mathcal{G}_n$.*

*Proof. Sketch.* We first draw a treewidth upper bound $k$ from an appropriate distribution. For example, a Poisson distribution with parameter $\lambda = \mathcal{O}(\log n / n)$ is sufficient. We have to ensure that each possible graph with treewidth up to $k$ gets a nonzero probability of being drawn. For that we first draw a $k$-tree, a maximal graph of treewidth $k$, and then take a random subgraph of it. $\qquad \square$

Note that we do not require that the patterns are sampled uniformly at random. It merely suffices that each pattern has a nonzero probability of being drawn. To satisfy a runtime of $\mathcal{O}(|V(G)|^{d+1})$ in expectation, for example, a Poisson distribution with $\lambda \leq \frac{1+d\log n}{n}$ is sufficient.

---

[3]Actually, it is sufficient to go up to the size of the second largest graph.

[4]Note that it does not affect the expressiveness results if we sample a pattern multiple times.

**Table 1:** Cross-validation accuracies on benchmark datasets

| method | MUTAG | IMDB-BIN | IMDB-MULTI | PAULUS25 | CSL |
|---|---|---|---|---|---|
| GHC-tree(6) | $89.28 \pm 8.26$ | $72.10 \pm 2.62$ | $48.60 \pm 4.40$ | $7.14 \pm 0.00$ | $10.00 \pm 0.00$ |
| GHC-cycle(8) | $87.81 \pm 7.46$ | $70.93 \pm 4.54$ | $47.41 \pm 3.67$ | $7.14 \pm 0.00$ | $100.00 \pm 0.00$ |
| GNTK | $89.46 \pm 7.03$ | $75.61 \pm 3.98$ | $51.91 \pm 3.56$ | $7.14 \pm 0.00$ | $10.00 \pm 0.00$ |
| GIN | $89.40 \pm 5.60$ | $70.70 \pm 1.10$ | $43.20 \pm 2.00$ | $7.14 \pm 0.00$ | $10.00 \pm 0.00$ |
| WL-kernel | $90.40 \pm 5.70$ | $73.12 \pm 0.40$ | - | $7.14 \pm 0.00$ | $10.00 \pm 0.00$ |
| ours (SVM) | $87.94 \pm 0.01$ | $70.37 \pm 0.01$ | $47.34 \pm 0.01$ | $100.00 \pm 0.00$ | $37.33 \pm 0.10$ |
| ours (MLP) | $88.55 \pm 0.01$ | $70.81 \pm 0.01$ | $48.29 \pm 0.01$ | $40.524 \pm 0.01$ | $13.27 \pm 0.01$ |

## 4 Related Work

The $k$-dimensional Weisfeiler-Leman (WL) test[5] [Cai et al., 1992] and the Lovász vector restricted to patterns up to treewidth $k$ are equally expressive [Dell et al., 2018; Dvořák, 2010]. We propose an efficiently computable embedding matching the expressiveness of $k$-WL, and hence also MPNNs and $k$-GNNs [Morris et al., 2019; Xu et al., 2019], in expectation, see Appendix D.

Dell et al. [2018] proposed a complete graph kernel based on homomorphism counts related to our $k_{\min}$ kernel. Instead of implicitly restricting the embedding to only a finite number of patterns, as we do, they weigh the homomorphism counts such that the inner product defined on the whole Lovász vectors converges. However, Dell et al. [2018] do not discuss how to compute their kernel and so, our approach can be seen as an efficient sampling-based alternative to their theoretical weighted kernel.

Using graph homomorphism counts as a feature embedding for graph learning tasks was proposed by Hoang and Maehara [2020] and Kühner [2021]. Hoang and Maehara [2020] discuss various aspects of homomorphism counts important for learning tasks, in particular, universality aspects and their power to capture certain properties of graphs, such as bipartiteness. Instead of relying on sampling patterns, which we use to guarantee expectation in completeness, they propose to use a fixed number of small pattern graphs. This limits the practical usage of their approach due to computational complexity reasons. In their experiments the authors only use tree (GHC-tree(6)) and cycle patterns (GHC-cycle(8)) up to size 6 and 8, respectively, whereas we allow patterns of arbitrary size and treewidth, guaranteeing polynomial runtime in expectation. Simiarly to Hoang and Maehara [2020], we use the computed embeddings as features for a kernel SVM (with RBF kernel) and an MLP.

Instead of embedding the whole graph into a vector of homomorphism counts, Barceló et al. [2021] proposed to use rooted homomorphism counts as node features in conjunction with a graph neural network (GNN). They discuss the required patterns to be as or more expressive than the $k$-WL test. We achieve this in expectation when selecting an appropriate sampling distribution, see Appendix D.

Wu et al. [2019] adapted random Fourier features [Rahimi and Recht, 2007] to graphs and proposed a sampling-based variant of the global alignment graph kernel. Similar sampling-based ideas were discussed before for the graphlet kernel [Shervashidze et al., 2009] and frequent-subtree kernels [Welke et al., 2015]. All three papers do not discuss expressiveness aspects, however.

## 5 Experiments

We performed some preliminary experiments on some benchmark datasets. To this end, we sample a fixed number $\ell = 30$ of patterns as described in Appendix A and compute the sampled min kernel as described in Section 3. Table 1 shows averaged accuracies of SVM and MLP classifiers trained on our feature sets. We follow the experimental design of Hoang and Maehara [2020] and compare to their published results. Even with as little as 30 features, the results of our approach are comparable to the competitors on real world datasets. Furthermore, it is interesting to note that a SVM with RBF kernel and our features performs perfectly on the PAULUS25 dataset, i.e., it is able to decide isomorphism for the strongly regular graphs in this dataset. It also shows good performance, although with high deviation, on the CSL dataset, where only the method specifically designed for this dataset, GHC-cycle, performs well. We also included GNTK [Du et al., 2019], GIN [Xu et al., 2019], and the WL-kernel [Shervashidze et al., 2011]. Baselines' accuracies are from Hoang and Maehara [2020].

---

[5]Note that this refers to the *folklore* $k$-WL test, also called $k$-FWL.

## 6 Conclusion

As future work, we will investigate approximate counts to make our implementation more efficient [Beaujean et al., 2021]. It is unclear how this affects expressiveness, as we loose permutation-invariance. Going beyond expressiveness results, our goal is to further study graph similarities suitable for graph learning, such as the cut distance as proposed by Grohe [2020]. Finally, instead of sampling patterns from a fixed distribution, a more promising variant is to adapt the sampling process in a sample-dependent manner. One could, for example, draw new patterns until each graph in the sample has a unique embedding (up to isomorphism) or at least until we can distinguish 1-WL classes. Alternatively, we could pre-compute frequent or interesting patterns as proposed by Schulz et al. [2018] and use them to adapt the distribution. Such approaches would use the power of randomisation to select an appropriate graph embedding in a data-driven manner, instead of relying on a finite set of fixed and pre-determined patterns like previous work [Barceló et al., 2021; Bouritsas et al., 2022].

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

**Table 2:** Preliminary runtime results in seconds of the embedding computation.

| method | MUTAG | IMDB-BIN | IMDB-MULTI | PAULUS25 | CSL |
|---|---|---|---|---|---|
| GHC-full(6) | $0.18 \pm 0.01$ | $11.28 \pm 0.24$ | $11.19 \pm 0.2$ | $2.92 \pm 0.049$ | $0.75 \pm 0.015$ |
| ours | $43.21 \pm 18.78$ | $516.85 \pm 301.37$ | $429.08 \pm 281.35$ | $351.26 \pm 877.9$ | $262.1 \pm 391.51$ |

## A  Sampling Details

Given a pattern size $N \in \mathbb{N}$, we first draw a treewidth upper bound $k < N$ given from some distribution. Then we want to sample any graph with treewidth at most $k$ with a nonzero probability. A natural strategy is to first sample a $k$-*tree*, which is a maximal graph with treewidth $k$, and then take a random subgraph of it. Uniform sampling of $k$-trees is described by Nie et al. [2015] and Caminiti et al. [2010]. Alternatively, the strategy of Yoo et al. [2020] is also possible. Note that we only have to guarantee that each pattern has a nonzero probability of being sampled; it does not have to be uniform. While guaranteed uniform sampling would be preferable, we resort to a simple sampling scheme that is easy to implement. We achieve a nonzero probability for each pattern of at most a given treewidth $k$ by first constructing a random $k$-tree $P$ through its tree decomposition, by uniformly drawing a tree $T$ on $N - k$ vertices and choosing a root. We then create $P$ as the (unique up to isomorphism) $k$-tree that has $T$ as tree decomposition. We then randomly remove edges from that $k$-tree i.i.d. with fixed probability (currently set to $0.1$). This ensures that each subgraph of $P$ will be created with nonzero probability.

## B  Implementation Details and Benchmark Datasets

Our source code is available on github[6] and the datasets in the correct format can be downloaded from a google drive[7]. We rely on the C++ code of Curticapean et al. [2017][8] to efficiently compute homomorphism counts. While the code computes a tree decomposition itself we decided to simply provide it with our tree decomposition of the $k$-tree which we compute anyway, to make the computation more efficient. Additionally, we use the cross-validation-based eveluation with SVM and MLP of Hoang and Maehara [2020][9].

MUTAG, IMDB-BIN, and IMDB-MULTI are taken from TU datasets [Morris et al., 2020], PAULUS25 is from Hoang and Maehara [2020][10], and CSL is from Murphy et al. [2019].

Table 2 shows preliminary runtime results. It reports the time taken for sampling 30 patterns and computing homomorphism counts for all patterns and all graphs in each database, i.e. the time to compute graph embeddings. We compare to the runtime of the combined GHC-full(6) = GHC-cycle(6) + GHC-tree(6) embedding computation. GHC is much faster, as GHC-full(6) only computes counts for a total of 18 tree and cycle patterns with up to six vertices. Our method allows for larger and more complex patterns and we pay a corresponding runtime cost. All runtimes were obtained on an AMD Ryzen 9 3900X 12-Core Processor with 64GB of RAM running Ubuntu 20.04. Note that our implementation is single-threaded and not yet well optimized.

## C  Proofs

**Theorem 5.** *Let $\mathcal{D}$ be a distribution on $\mathcal{G}_n$ with full support and $G \in \mathcal{G}_n$. Then the graph embedding $\varphi_F(G) = \hom(F, G)e_F$ with $F \sim \mathcal{D}$ and the corresponding kernel $k$ are complete in expectation.*

*Proof.* Let $\mathcal{D}$ and $\varphi_F$ with $F \sim \mathcal{D}$ as stated and $G \in \mathcal{G}_n$. Then

$$g = \mathbb{E}_F[\varphi_F(G)] = \sum_{F' \in \mathcal{G}_n} \Pr(F = F') \hom(F', G)e_{F'}.$$

The vector $g$ has the entries $(g)_{F'} = \Pr(F = F') \hom(F', G)$. Let $G'$ be a graph that is non-isomorphic to $G$ and let $g' = \mathbb{E}_F[\varphi_F(G')]$ accordingly. By Theorem 1 we know that $\hom(\mathcal{G}_n, G) \neq$

---

[6]https://github.com/pwelke/homcount
[7]https://drive.google.com/file/d/1aBwlk-9qXOSBKDRFEkXstcsWVxfPn0z0
[8]https://github.com/ChristianLebeda/HomSub
[9]https://github.com/gear/graph-homomorphism-network
[10]originally from https://www.distanceregular.org/graphs/paulus25.html

$\hom(\mathcal{G}_n, G')$. Thus, there is an $F'$ such that $\hom(F', G) \neq \hom(F', G')$. By definition of $\mathcal{D}$ we have that $\Pr(F = F') > 0$ and hence $\Pr(F = F') \hom(F', G) \neq \Pr(F = F') \hom(F', G')$ which implies $g \neq g'$. That shows that $\mathbb{E}_F[\varphi_F(\cdot)]$ is complete and concludes the proof. $\qquad\square$

**Lemma 6.** $k_{\min}$ *is a complete kernel on $\mathcal{G}_\infty$.*

*Proof.* Let $G, H \in \mathcal{G}_\infty$. We have to show that

$$\varphi_{\tilde\infty}(G) = \varphi_{\tilde\infty}(H) \Leftrightarrow G \simeq H \;,$$

where $G \simeq H$ indicates that $G$ and $H$ are isomorphic. There are two cases:

$|V(G)| = |V(H)|$: Then, by Theorem 1 we have $\varphi_N(G) = \varphi_n(H)$ iff $G \simeq H$ for $N = \min\{|V(G)|, |V(H)|\} = |V(G)| = |V(H)|$.

$|V(G)| \neq |V(H)|$: Let w.l.o.g. $0 < |V(G)| < |V(H)|$. Let $P$ be the graph on exactly one vertex. Then $\hom(P, G) < \hom(P, H)$, i.e., we can distinguish graphs on different numbers of vertices using homomorphism counts. As $\min\{|V(G)|, |V(H)|\} \geq 1$, we have $P \in \mathcal{G}^{|V(G)|}$ and hence $\varphi_{|V(G)|}(G) \neq \varphi_{|V(G)|}(H)$. The other direction follows directly from the fact that homomorphism counts are invariant under isomorphism. $\qquad\square$

**Theorem 7.** *Let $\mathcal{D}$ be a distribution on $\mathcal{G}_\infty$ with full support and $G \in \mathcal{G}_\infty$. Then $\bar\varphi_F(G) = \hom_{|V(G)|}(F, G)e_F$ with $F \sim \mathcal{D}$ and the corresponding kernel are complete in expectation.*

*Proof.* We can apply the same arguments as before from Theorem 5 to show that the expected embeddings of two graphs $G, H$ with size $n' = \min\{|V(G)|, |V(H)|\}$ are equal iff their Lovász vector restricted to size $n'$ are equal. By Lemma 6 we know that the latter only can happen if the two graphs are isomorphic. $\qquad\square$

**Proposition 8.** *There exists no distribution $\mathcal{D}$ with full support on $\mathcal{G}_\infty$ such that the expected runtime of Eq. (1) becomes polynomial in $|V(G)|$ for all $G \in \mathcal{G}_\infty$.*

*Proof.* Let $\mathcal{D}$ be such a distribution and let $\mathcal{D}'$ be the marginal distribution on the treewidths of the graphs given by $p_k = \Pr_{F \sim \mathcal{D}}(\mathrm{tw}(F) = k) > 0$. Let $G$ be a given input graph in the sample with $n = |V(G)|$. Díaz et al. [2002] has shown that computing $\hom(F, G)$ takes time $\mathcal{O}\left(|V(F)||V(G)|^{\mathrm{tw}(F)+1}\right)$. Assume for the purpose of contradiction that we can guarantee an expected polynomial runtime (ignoring the $|V(F)|$ and constant factors for simplicity):

$$\mathbb{E}_{F \sim \mathcal{D}}[n^{\mathrm{tw}(F)+1}] = \sum_{k=1}^{\infty} p_k n^{k+1} \leq Cn^c$$

for some constants $C, c \in \mathbb{N}$. Then for all $k \geq c$, it must hold that $p_k n^{k+1} \leq Cn^c$, as all summands are positive. However, for large enough $n$ the left hand side is larger than the right hand side. Contradiction. $\qquad\square$

**Theorem 9.** *There exists a distribution $\mathcal{D}$ such that computing the expectation complete graph embedding $\bar\varphi_X(G)$ takes polynomial time in $|V(G)|$ in expectation for all $G \in \mathcal{G}_n$.*

*Proof.* Let $G \in \mathcal{G}_n$. Draw a treewidth upper bound $k$ from a Poisson distribution with parameter $\lambda$ to be determined later. Select a distribution $\mathcal{D}_{n,k}$ which has full support on all graphs with treewidth up to $k$ and size up to $n$, for example, the one described in Appendix A. Using the algorithm of [Díaz et al., 2002] this gives, for some constant $C \in \mathbb{N}$, an expected runtime of

$$\begin{aligned} \mathbb{E}_{k \sim \mathrm{Poi}(\lambda), F \sim \mathcal{D}_{n,k}} \left[ C|V(F)||V(G)|^{\mathrm{tw}(F)+1} \right] &\leq \mathbb{E}_{k \sim \mathrm{Poi}(\lambda)} \left[ Cn^{k+2} \right] \\ &= \sum_{k=0}^{\infty} \frac{\lambda^k e^{-\lambda}}{k!} Cn^{k+2} = \frac{Cn^2}{e^\lambda} e^{\lambda n}. \end{aligned}$$

We need to bound the right hand side by some polynomial $Dn^d$ for some constants $D, d \in \mathbb{N}$. By rearranging terms we see that

$$\lambda \leq \frac{\ln \frac{D}{C} + (d-2) \ln n}{n - 1} = \mathcal{O}\left(\frac{\log n}{n}\right)$$

is sufficient.

$\square$

## D   Matching the Expressiveness of $k$-WL in Expectation

We devise a graph embedding matching the expressiveness of the $k$-WL test in expecation.

**Theorem 10.** *Let $\mathcal{D}$ be a distribution with full support on the set of graphs with treewidth up to $k$. The resulting graph embedding $\varphi_F^{k\text{-WL}}(\cdot)$ with $F \sim \mathcal{D}$ has the same expressiveness as the $k$-WL test in expectation. Furthermore, there is a specific such distribution such that we can compute $\varphi_F^{k\text{-WL}}(G)$ in expected polynomial time $\mathcal{O}(|V(G)|^{k+1})$ for all $G \in \mathcal{G}_\infty$.*

*Proof.* Let $\mathcal{T}_k$ be the set of graphs with treewidth up to $k$ and $\mathcal{D}$ be a distribution with full support on $\mathcal{T}_k$. Then by the same arguments as before in Theorem 5, the expected embeddings of two graphs $G$ and $H$ are equal iff their Lovász vectors restricted to patterns in $\mathcal{T}_k$ are equal. By Dvořák [2010] and Dell et al. [2018] the latter happens iff $k$-WL returns the same color histogram for both graphs. This proves the first claim.

For the second claim note that the worst-case runtime for any pattern $F \in \mathcal{T}_k$ is $\mathcal{O}\left(|V(F)||V(G)|^{k+1}\right)$ by Díaz et al. [2002]. However, the equivalence between homomorphism counts on $\mathcal{T}_k$ and $k$-WL requires to inspect also patterns $F$ of all sizes, in particular, also larger than the size $n$ of the input graph. To remedy this, we can draw the pattern size $m = |V(F)|$ from some distribution with bounded expectation and full support on $\mathbb{N}$. For example, the geometric $m \sim \text{Geom}(p)$ with any constant parameter $p \in (0, 1)$ and expectation $\mathbb{E}[m] = \frac{1}{1-p}$ is sufficient. By linearity of expectation then

$$\mathbb{E}_{F \sim \mathcal{D}}\left[|V(F)||V(G)|^{\text{tw}(F)+1}\right] \leq \mathbb{E}_{F \sim \mathcal{D}}\left[|V(F)||V(G)|^{k+1}\right]$$
$$= \mathbb{E}_{F \sim \mathcal{D}}\left[|V(F)|\right]|V(G)|^{k+1}$$
$$= \mathcal{O}\left(|V(G)|^{k+1}\right).$$

$\square$

Note that for the embedding $\varphi_F^{k\text{-WL}}(\cdot)$ Lemma 8 does not apply. In particular, the used distribution guaranteeing polynomial expected runtime is independent of $n$ and can be used for all $\mathcal{G}_\infty$.

