# OpenReview forum: "Expectation Complete Graph Representations using Graph Homomorphisms"
_logconference.io/LOG/2022/Conference — LoG 2022 Poster_

### Official Review · Reviewer_SVZ3 · 2022-10-03

**Overall Score:** 6
**Confidence:** 4

**Review:**

**Summary** - The authors propose an interesting solution to learn graph embeddings that are maximally expressive in expectation. Contrary to the previous methods that either focus on high expressiveness or high computational efficiency, this work tries to bridge the two properties with reasonable tradeoffs. While the comparison of Lovasz vector allows completeness, the authors study a relaxed property of *completeness in expectation*. To further define a kernel on $\mathcal{G}_{\infty}$, the authors compute an amortized vector of homomorphism counts and show that it is complete. They also provide a solution to compute the embedding vector for a graph in polynomial time with justification in theorem 9. Experiments conducted on 5 benchmarks and comparison against 4 baselines suggest that the proposed approach is promising.

**Strengths and Weeknesses** -
1. The idea proposed in this work is interesting with sufficient justification from the literature.
2. The proofs are simple and sound.
3. The experimental results in table 1 are promising, although the method doesn’t outperform the baselines except on the PAULUS25 dataset.
4. The write up of the paper has not been thoroughly verified. For example the overloading of the symbol $\varphi$ to denote homomorphism in line 49 as well as to denote the embedding function in definition 3 line 61 is confusing. I suggest the authors to reparse the paper.
5. I believe the related work needs to be elaborated more thoroughly. Detailed comparison against the works of “Dell et al. [2018]” and “Hoang and Maehara [2020]” is needed. This work is fairly similar to the aforementioned.
6. The emphasis on runtime efficiency is strong. Despite this, the computation of the count of homomorphisms is expensive. Some experiments on the runtime analysis of different methods will be very useful.

Although there are some concerns surrounding the work as mentioned previously, I deviate to acceptance of the work.

---

### Official Review · Reviewer_AuRT · 2022-10-20

**Overall Score:** 6
**Confidence:** 3

**Review:**

**Summary**: In this paper, the authors leverage Lovász’ characterization of graph isomorphism to develop a new kind of graph embedding that in expectation is able to distinguish all non-isomorphism graphs. First, they provide analyses on the completeness of the proposed randomized graph embedding on the graph isomorphism problem. Then, the authors propose a sampling strategy to efficiently compute the graph embedding. Experiments on TU datasets and synthetic datasets are conducted to verify the validity of the proposed graph embedding.

**+ves**:

The idea of using randomized graph homomorphism counts as graph embeddings is interesting.

**Concerns**:

**Regarding the gap between theoretical expressive power and empirical implementation of the proposed graph embedding**: The proposed graph embedding can achieve the completeness on graph isomorphism problem in expectation. However, the practical implementation needs to design sophisticated sampling strategies to obtain an empirical estimation of such embedding expectation. The authors should provide both analyses on the sample complexity of the randomized graph embedding and empirical ablation studies on the sampling strategies including pattern graph distribution, the number of samples, treewidth $k$, and other factors influencing the graph embedding. Otherwise, readers would be confused since there exists indeed a gap between theoretical expressive power and empirical implementation of the proposed graph embedding.

**Regarding the experimental results**: First, the comparison of baselines is not comprehensive enough. Note that the datasets used in this work are popularly used in graph machine learning. At least Classic Message Passing Neural Networks (MPNNs) should also be included to compare with the proposed graph embedding. Second, the performance on most datasets is not superior to that of baselines, even for simple MPNNs. Thus, the results can hardly demonstrate the validity of the proposed graph embedding.

**Overall**, if the authors can address my above concerns well, I would like to increase my scores accordingly.

******** After Rebuttal ********
I lean towards acceptance. The authors should consider to include the above advice in the full version of this paper.

---

### Official Review · Reviewer_HNLR · 2022-10-20

**Overall Score:** 8
**Confidence:** 3

**Review:**

# Summary

The paper proposes a new probabilistic method based on graph homomorphism counts, that is able to distinguish any two non-isomorphic graphs *in expectation*. In contrast to deterministic approaches for graph isomorphism testing (e.g. the Weisfeiler Leman test), this probabilistic relaxation of the problem provides a new way to keep the computational complexity control, while being maximally expressive.

# Strengths

- The limitations of the Weisfeiler-Lehman hierarchy have been intensively scrutinised in recent years and the proposed approach represents a nice and radically different alternative for studying the expressive power of graph-based models and for designing new ones. In particular, the proposed approach based on stochastic graph homomorphism covers two settings that are not covered by WL: (1) Models that are *not* based on message passing and (2) stochastic models.
- The paper makes a nice transition from graphs of a certain size to the set of all finite graphs, while carefully addressing the mathematical and computational issues that emerge as a consequence.

# Weaknesses

- The proposed approach does not (directly) make use of the graph features as homomorphism counts are purely based on the graph structure.
- Perhaps related to the above, the method seems to perform well only on synthetic datasets without features, but struggles on real-world datasets.
- I think a useful experiment (currently missing) would be to analyse how the performance of the model evolves as a function of the number of patterns. I expect that bigger graphs require a higher number of samples. If the number of samples becomes reasonably high (e.g. 100), the computational load on very large graphs could become impractical.

# Questions and Suggestions

- Are there any ways that graph features could be integrated into your framework? How are the node features on the real-world datasets in Table 1 used? Are they simply supplied as additional input to the SVM / MLP?
- I think it would be useful to add in the appendix the pseudocode for the model to make it more clear how it works.
- In Def 4, there is a notation clash since $X$ was used to denote vector spaces in Def 3.
- Line 52: $x \in \mathcal{X}$ rather than $x \in X$.

---

### Official Review · Reviewer_kHM5 · 2022-10-22

**Overall Score:** 6
**Confidence:** 3

**Review:**

Summary:

This work is focused on graph embedding methods with strong expressive power. Based on previous expressive but impractical method, this work proposes a sampling-based alternatives while maintaining expressitivity in expectation. The proposed method is evaluated via experiments with several standard GNN baselines.

Pros:
1. The discussion of related work is comprehensive, which clarifies the position of this work.
2. The definitions and development of theory are clear.
3. The proposed method is natural, and it is verified with necessary experiments.

Cons:
1. The runtime of the proposed method versus baselines is not presented. Additionally it would be great to show results of (maybe truncated) previous complete graph embedding method to show whether there is any significant tradeoff between efficiency and expressitivity.
2. It would be great to add some discussions about the application scenario of such a method, for instance what is its advantage on a specific kind of datasets. My naive understanding is that 1) for social networks, it is very expensive to perform any (local) high-order counting, and 2) for biological data, handcrafted substructures/patterns seem well-founded from prior knowledge. Please feel free to correct me.

Small suggestions:
1. The reference to CSL dataset is missing. I suppose it is from Murphy et al.

Final comment:

I would like to suggest an acceptance as a short abstract.

---

### Meta-Review · Area_Chair_n1y5 · 2022-11-09

**Confidence:** 4
**Recommendation:** Accept

**Meta Review:**

The paper studies graph embedding approaches via the well-known homomorphism characterization of the graph isomorphism problem, and develops sampling-based approximation. Some reviewers criticize the lack of discussion of related work, claiming that the present work is not entirely novel, or criticize the usefulness of the method on real-world datasets. However, all reviewers agree that the paper should be accepted.

---

### Decision · Program_Chairs · 2022-11-23

Accept (Poster)